# Limited window for donation of convalescent plasma with high live-virus neutralizing antibody titers for COVID-19 immunotherapy

Abhinay Gontu[1,11], Sreenidhi Srinivasan [2,11], Eric Salazar[3,11], Meera Surendran Nair[1], Ruth H. Nissly [1], Denver Greenawalt[1], Ian M. Bird[1], Catherine M. Herzog[2], Matthew J. Ferrari [2,4], Indira Poojary[2], Robab Katani[2], Scott E. Lindner[2,5], Allen M. Minns[2,5], Randall Rossi[2], Paul A. Christensen[3], Brian Castillo[3], Jian Chen[3], Todd N. Eagar[3,6], Xin Yi[3,6], Picheng Zhao[3], Christopher Leveque[3], Randall J. Olsen[3,6,7], David W. Bernard[3,7], Jimmy Gollihar[3,8], Suresh V. Kuchipudi[1,9 ✉], James M. Musser[3,5,6 ✉] & Vivek Kapur[2,9,10 ✉]

Millions of individuals who have recovered from SARS-CoV-2 infection may be eligible to participate in convalescent plasma donor programs, yet the optimal window for donating high neutralizing titer convalescent plasma for COVID-19 immunotherapy remains unknown. Here we studied the response trajectories of antibodies directed to the SARS-CoV-2 surface spike glycoprotein and in vitro SARS-CoV-2 live virus neutralizing titers (VN) in 175 convalescent donors longitudinally sampled for up to 142 days post onset of symptoms (DPO). We observed robust IgM, IgG, and viral neutralization responses to SARS-CoV-2 that persist, in the aggregate, for at least 100 DPO. However, there is a notable decline in VN titers ≥160 for convalescent plasma therapy, starting 60 DPO. The results also show that individuals 30 years of age or younger have significantly lower VN, IgG and IgM antibody titers than those in the older age groups; and individuals with greater disease severity also have significantly higher IgM and IgG antibody titers. Taken together, these findings define the optimal window for donating convalescent plasma useful for immunotherapy of COVID-19 patients and reveal important predictors of an ideal plasma donor.

[1] Department of Veterinary and Biomedical Sciences, Pennsylvania State University, University Park, PA, USA. [2] Huck Institutes of the Life Sciences, Pennsylvania State University, University Park, PA, USA. [3] Department of Pathology and Genomic Medicine, Houston Methodist Hospital, Houston, TX, USA. [4] Department of Biology, Pennsylvania State University, University Park, PA, USA. [5] Department of Biochemistry, Microbiology, and Molecular Biology, Pennsylvania State University, University Park, PA, USA. [6] Department of Pathology and Laboratory Medicine, Weill Cornell Medical College, New York, NY, USA. [7] Center for Molecular and Translational Human Infectious Diseases, Houston Methodist Research Institute, Houston, TX, USA. [8] CCDC Army Research Laboratory-South, Austin, TX, USA. [9] Center for Infectious Disease Dynamics, Pennsylvania State University, University Park, PA, USA. [10] Department of Animal Science, Pennsylvania State University, University Park, PA, USA. [11] These authors contributed equally: Abhinay Gontu, Sreenidhi Srinivasan, Eric Salazar. ✉email: skuchipudi@psu.edu; jmmusser@houstonmethodist.org; vkapur@psu.edu

The kinetics and longevity of the antibody response to severe acute respiratory syndrome coronavirus 2 (SARS-CoV-2) are poorly understood. This knowledge is essential for determining if individuals have been infected, elucidating host and virus factors that influence the magnitude and persistence of serological responses, assessing whether an individual is sufficiently protected from re-infection, and evaluating the effectiveness of vaccination strategies to contain the pandemic[1]. Additionally, understanding antibody kinetics and persistence is essential to determine correlates of live-virus neutralization (VN) titers required for qualifying donors of convalescent plasma for use in immunotherapy[2–5]. These questions are especially important given (1) the mounting interest in SARS-CoV-2 vaccine research (2) the rising use of convalescent plasma, which was recently granted Emergency Use Authorization by the Food and Drug Administration to treat COVID-19 patients[6,7], and (3) emerging evidence that transfusion of anti-Spike receptor binding domain (S/RBD) IgG ≥1350 titer plasma within 72 h (h) of hospitalization significantly improves survival and health outcomes[8,9].

Antibodies directed to the SARS-CoV-2 surface spike glycoprotein (S) ectodomain (S/ECD) and receptor-binding domain (S/RBD) neutralize SARS-CoV-2 in vitro, and their titers can serve as effective surrogates for virus neutralization (VN)[8–11]. These titers have also been used to identify suitable convalescent plasma donors for COVID-19 immunotherapy[11–13]. However, there is considerable uncertainty about the robustness and persistence of the serological responses to SARS-CoV-2. Some reports suggest variable duration and resilience of serum IgG or IgM antibodies to S or other viral proteins[10,11,14,15], whereas others report that serological and neutralizing responses begin to wane and approach undetectable levels within weeks after infection[11,14,16,17]. As a consequence, the optimal time window for convalescent plasma donation for COVID-19 immunotherapy remains unknown, as are the defining characteristics of individuals who might represent suitable donors for convalescent plasma.

To better understand the kinetics of the serological response to SARS-CoV-2, we determined the temporal profiles of IgM, IgG, and VN responses in a cohort of 175 convalescent plasma donors, including 105 who had donated multiple times. Plasma samples ($n = 540$) were collected up to 142 days after the onset of the donors' first symptoms (days post-symptom onset (DPO); Table 1, Supplementary Table S1). We used a Fab fragment-based assay to assess total antibody titers against S/ECD and S/RBD, an isotype-specific assay to measure anti-S/RBD IgM and IgG titers, and a live-virus assay to determine SARS-CoV-2 VN titers[12]. We identify a robust and persistent live virus VN and serological response to SARS-CoV-2 S/ECD and S/RBD but conclude there is a limited donation window of ~60 DPO for high-titer anti-spike protein convalescent plasma suitable for immunotherapy in COVID-19 patients.

## Results

**Distribution, correlation, and trajectories of antibody titers against SARS-CoV-2.** We discovered robust IgM, IgG, and VN responses in the majority of individuals, with moderate to strong correlation regardless of assay type (Fig. 1A, B). Only 4 of 175 (2.3%; 95% confidence interval (CI): 0.9–5.7%) individuals had undetectable levels of IgG, IgM, or total antibody to S/RBD or S/ECD at initial sampling, whereas a significantly higher fraction (29 of 114; 25.4%; 95% CI: 18.3–34.1%) had undetectable VN titers ($z$-score = 6; $P < 0.01$).

We next determined the patterns of distribution of IgM and IgG background-corrected optical density (OD) values and titers

**Table 1 Demographics and characteristics of the plasma donor cohort.**

| Patient characteristics | Samples, n (%) | Individuals, n (%) |
|---|---|---|
| *Sex* | | |
| Female | 213 (39.4) | 88 (50.3) |
| Male | 327 (60.6) | 87 (49.7) |
| *Age* | | |
| 20–30 | 95 (17.6) | 26 (14.9) |
| 31–40 | 117 (21.7) | 39 (22.3) |
| 41–50 | 166 (30.7) | 51 (29.1) |
| 51–60 | 117 (21.7) | 40 (22.9) |
| >60 | 45 (8.3) | 19 (10.9) |
| Average (95% CI) | 43.8 (42.7–44.9) | 44.9 (43.0–46.8) |
| Median (IQR) | 44 (33–53) | 46 (36–54) |
| Range | 20–78 | 20–78 |
| *Severity* | | |
| 1 | 244 (45.2) | 76 (43.4) |
| 2 | 182 (33.7) | 63 (36.0) |
| 3 | 23 (4.3) | 10 (5.7) |
| 4 | 44 (8.1) | 15 (8.6) |
| 5 | 47 (8.7) | 11 (6.3) |
| Median (IQR) | 2 (1–2) | 2 (1–2) |
| Range | 1–5 | 1–5 |
| *Dyspnea* | | |
| No | 250 (46.3) | 79 (45.1) |
| Yes | 290 (53.7) | 96 (54.9) |
| *DPO* | | |
| <31 | 39 (7.2) | 35 (20.0) |
| 31–60 | 181 (33.5) | 89 (50.9) |
| 61–90 | 173 (32.0) | 44 (25.1) |
| 91–120 | 122 (22.6) | 7 (4.0) |
| >120 | 25 (4.6) | – |
| Average (95% CI) | 70.8 (68.4–73.3) | 49.5 (46.5–52.5) |
| Median (IQR) | 68 (48–93) | 46 (32–63) |
| Range | 17–142 | 17–108 |
| *Hospitalization* | | |
| No | 428 (79.3) | 141 (80.6) |
| Yes | 112 (20.7) | 34 (19.4) |
| Total | 540 | 175 |

Study cohort characteristics such as sex, age, disease severity, dyspnea, and hospitalization records are provided.

over time (Fig. 1C–F). Titers peaked at approximately 30 DPO and persisted through 140 DPO (Fig. 1C–G), with the IgG titer consistently higher than the IgM titer. The titer ratios began to diverge after 60 DPO (Fig. 1D, F), but remained strongly correlated over the first 140 DPO (Pearson's $r = 0.71$; 95% CI: 0.67–0.75).

**Survival analysis of IgG, IgM, and VN antibody titers to SARS-CoV-2 spike-receptor binding domain (S/RBD).** To further study the trajectory of antibody persistence, we performed survival analyses on IgM and IgG titers on all 540 samples obtained from 175 individual donors (Fig. 2). Consistent with the temporal distribution of titers, survival analyses showed that the proportion of S/RBD IgG seropositive convalescent individuals remained high through 140 DPO (Fig. 2A, B). Our large and well-characterized convalescent plasma library with longitudinally donated samples also enabled detailed assessment of VN response persistence. The proportion of individuals with a VN titer ≥160 remained above 80% through the first 60 DPO but declined to <20% between DPO 61 and 120 (Fig. 2C, D).

We previously reported that an S/RBD ≥ 1350 titer serves as a robust marker for plasma donors with VN ≥ 160[12] (Supplementary Data 1). Here we confirm a high positive likelihood ratio (LR+; 13.43) for a VN ≥ 160 when S/RBD titers are ≥1350 early

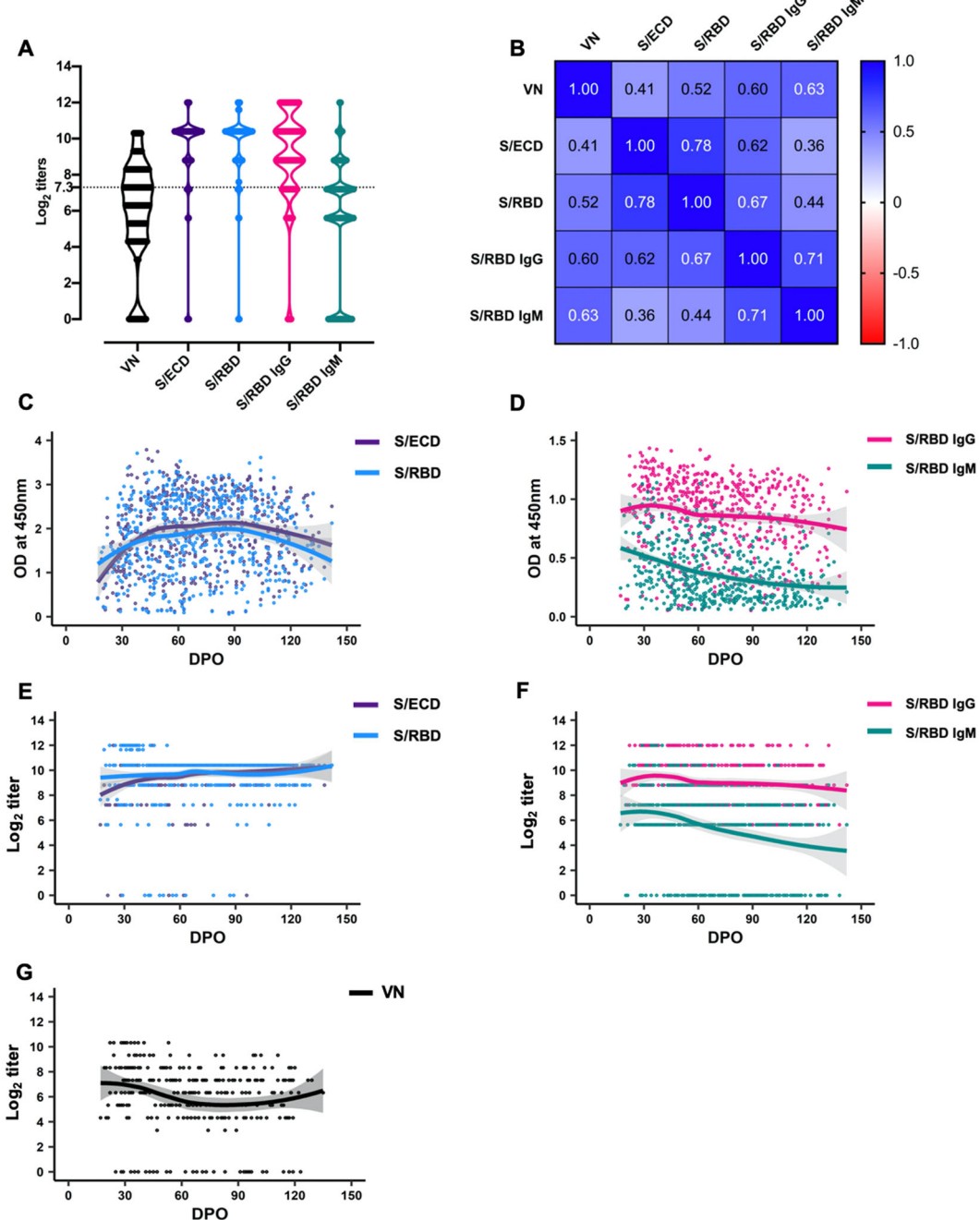

**Fig. 1 Distribution, correlation, and trajectories of antibody titers against SARS-CoV-2. A** Violin plots showing distribution of virus neutralization titers ($n = 305$); total antibody ($n = 538$), and specific isotype antibody IgG and IgM ($n = 540$) titers to SARS-CoV-2 spike-ectodomain (S/ECD) and spike-receptor binding domain (S/RBD) in convalescent plasma samples ($\log_2$ transformed values). The means of the distribution among the titers were significantly different, except between S/ECD and S/RBD [One-way ANOVA, Tukey's multiple comparison (mixed-effects model), $P < 0.05$]. The dashed line at $\log_2$ titer represents VN titer of 1:160. **B** Pairwise comparison of the assays showed a moderate to strong correlation between the total and isotype-specific IgG and IgM antibody estimates with virus neutralization assays. **C, D** Optical density (OD) (at 450 nm) for the indirect ELISAs indicating total or isotype-specific IgG and IgM antibody levels. **E, F** Titers of the total or isotype-specific IgG and IgM antibodies. The IgG and IgM titers peaked around 30 days post onset (DPO) of symptoms. High IgG titers persisted until 140 DPO, while IgM titers trended lower but persisted until 140 DPO. **G** Neutralizing antibody titers persisted until 140 DPO. A locally estimated scatterplot smoothing (LOESS) regression curve is fitted to the data.

(1–30 DPO) post onset of symptoms (Supplementary Data 1). Extended longitudinal analyses through 140 DPO showed that S/ECD and S/RBD ≥ 1350 persisted longer than VN ≥ 160, with significantly different survival curves ($P < 0.001$) for 1–140 DPO and overall LRs+ of 1.34 for S/ECD and 1.61 for S/RBD (Fig. 2C; Supplementary Fig. S7 and Supplementary Data 1). In contrast, S/RBD IgG ≥ 1350 appeared to be a reliable predictor of VN ≥ 160,

and S/RBD IgG ≥ 1350 survival was statistically indistinguishable from that of VN ≥ 160 (Fig. 2D), with an overall LR + of 3.18 and a negative likelihood ratio (LR-) of 0.26 (Supplementary Data 1).

We next investigated the survival and predictive values of S/RBD IgM ≥ 450 as compared to VN ≥ 160 (Fig. 2D, Supplementary Data 1). An S/RBD IgM titer ≥ 450 was selected because the magnitude of IgM response was approximately threefold lower

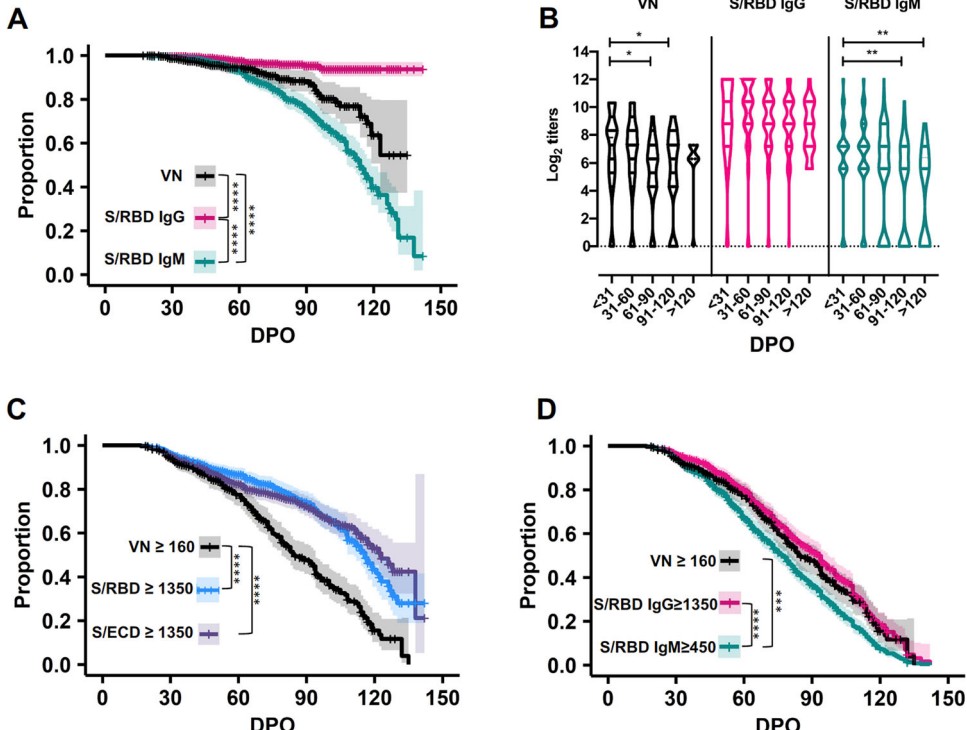

**Fig. 2 Survival analysis of IgG, IgM, and VN antibody titers to SARS-CoV-2 spike-receptor binding domain (S/RBD).** These data represent IgG and IgM antibody titers to SARS-CoV-2 spike-receptor binding domain (S/RBD) in 540 samples and virus neutralizing antibody (VN) titers in 305 samples collected from convalescent individuals ($n = 175$) during the first 140 days post onset of symptoms (DPO). **A** Proportion of S/RBD IgG seropositive convalescent individuals remained high through 140 DPO, while IgM seropositivity remained high through the first 60 DPO and then steadily declined over the next 60 days (log-rank test; ****$P < 0.0001$). The proportion of individuals with VN responses also began to decline 60 DPO, with ~50% of individuals remaining seropositive with VN test through 140 DPO (log-rank test; ***$P < 0.001$). **B** Violin plots showing a significant decline in VN and IgM titers with time (ordinary one-way ANOVA, Tukey's multiple comparison test; *$P < 0.05$; **$P < 0.01$); the IgG titers remained stable until after 120 DPO. Comparison of proportion of individuals seropositive with S/RBD, S/ECD, and S/RBD IgG titers ≥1350 as well as with S/RBD IgM titer ≥450 to the proportion of individuals possessing VN titers ≥160 through 140 DPO are depicted in **C**, **D**, respectively (***$P < 0.001$; ****$P < 0.0001$).

than that of IgG (Fig. 1F). The results showed that S/RBD IgM ≥450 had a similar survival profile to VN ≥160 but waned significantly faster ($P < 0.01$; Fig. 2D). While S/RBD IgM ≥450 had an overall LR+ of 3.72, it also had a LR- of 0.69, which would likely result in an unacceptable number of suitable donors with VN ≥160 being excluded. Together, these results indicate that S/RBD IgG ≥1350, but not IgM ≥450 or S/RBD or S/ECD total antibody ≥1350, is a suitable marker to identify plasma donors for COVID-19 immunotherapy.

**Distribution of antibody titers against SARS-CoV-2 based on age, severity scores, and presence of dyspnea.** We next tested the hypothesis that particular donor characteristics predicted a more robust serological and neutralization response. Consistent with the hypothesis, individuals 30 years of age or younger had significantly lower VN, IgG, and IgM antibody titers than those in the older age groups (Fig. 3A). Individuals between 20–30 years of age also had significantly faster decline in IgG ($P < 0.05$) and IgM ($P < 0.05$) than did those >60 years of age (Fig. 3B–D and Supplementary Fig. S4A). Consistent with recent evidence that disease severity correlates with the magnitude and duration of serological response[12,18,19], we found that individuals with disease severity scores of 4 or 5 on a 5-point disease severity scale had significantly higher IgM and IgG antibody titers than those with lower severity scores (Fig. 3E). In addition, survival analyses of IgG and IgM antibody titers revealed that individuals with mild/moderate symptoms scores of 1, 2, or 3 had significantly different survival curves for IgM ($P < 0.0001$) and VN ($P < 0.05$) than did

those with higher disease severity scores (Fig. 3F–H and Supplementary Fig. S4B). Notably, all individuals with high severity scores had detectable IgM at their last measurement point, as did all individuals who were >60 years of age. This may indicate confounding or interaction between age and disease severity affecting the magnitude and persistence of serological response. The rate of loss of IgM seropositivity to S/RBD was significantly higher for the youngest (20–30 years) compared to the oldest (>60 years) age groups (log-rank test, $P < 0.01$), and this effect remained significant when individuals with high severity scores were excluded. Age and severity score were only weakly correlated (Spearman rank correlation = 0.08; $P = 0.07$), but formal analysis of confounding or interactions between age and severity was precluded due to data frailty and requires further study. Regardless, our findings suggest that convalescent individuals <30 years of age and those with lower disease severity scores are less likely to be suitable donors of convalescent plasma for immunotherapy for COVID-19 patients than individuals in >30 age group with a history of more severe disease. Finally, the results show that individuals with dyspnea had significantly higher VN, IgG, and IgM and antibody titers than those who did not (Fig. 3I), and IgM seropositivity declined significantly faster in individuals with dyspnea (log-rank test, $P < 0.0001$) (Fig. 3J–L). Significant differences were observed between the S/RBD IgG titers of males ($n = 327$) and females ($n = 213$). There were no differences observed in the IgM and VN titers of the study population when stratified by sex. Distribution of antibody titers against SARS-CoV-2 based on sex is shown in Supplementary Fig. S3.

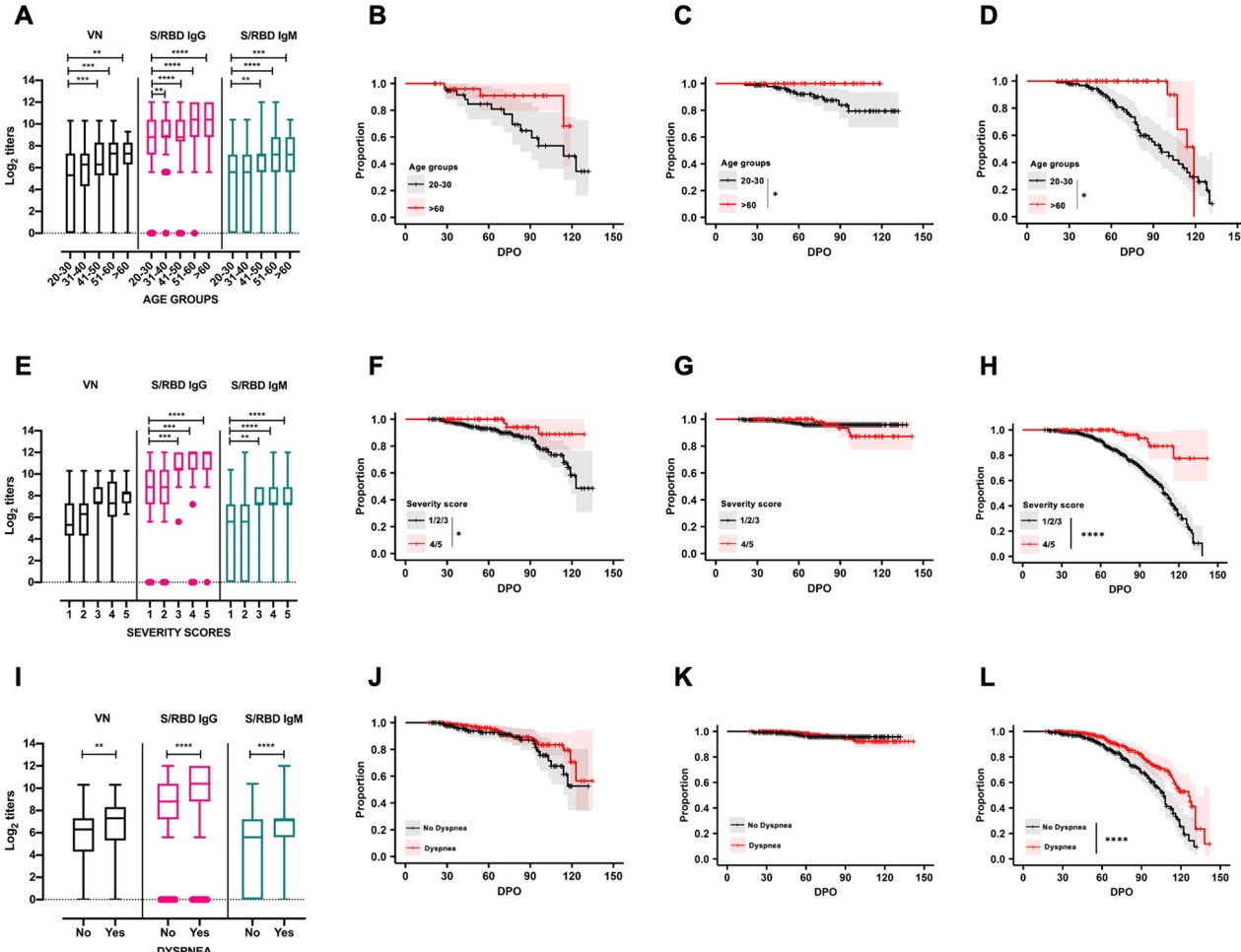

**Fig. 3 Distribution of antibody titers against SARS-CoV-2 based on age, severity scores, and presence of dyspnea.** These data represent samples collected from convalescent individuals ($n = 175$) during the first 140 days post-symptom onset (DPO). **A** Individuals <31 years of age have significantly lower IgG, IgM, and viral neutralizing antibody (VN) titers than those >40 years of age in this cohort (Ordinary one-way ANOVA, Tukey's multiple comparison test; **$P < 0.01$; ***$P < 0.001$; ****$P < 0.0001$). Survival analysis of (**B**) VN, (**C**) IgG, and (**D**) IgM antibody titers during the first 140 DPO in convalescent individuals within the age groups of 20–30 ($n = 95$ samples) and >60 ($n = 45$ samples) (log-rank test, $P > 0.05$ for VN antibodies, *$P < 0.05$ for IgG and IgM). **E** Individuals with a severity score of 1 have significantly lower IgM and IgG titers than those above a score of 3 (Ordinary one-way ANOVA, Tukey's multiple comparison test; **$P < 0.01$; ***$P < 0.001$; ****$P < 0.0001$). Survival analysis of (**F**) VN, (**G**) IgG, and (**H**) IgM antibody titers in relation to severity scores grouped as mild (1/2/3) and severe (4/5) in convalescent individuals during the first 140 DPO (log-rank test, *$P < 0.05$ for VN antibodies, $P > 0.05$ for IgG, ****$P < 0.0001$ for IgM). **I** Individuals with dyspnea had significantly higher VN, IgM, and IgG titers (unpaired $t$ test, two-tailed; **$P < 0.01$; ****$P < 0.0001$). Survival analysis of (**J**) VN, (**K**) IgG, and (**L**) IgM antibody titers in relation to occurrence of dyspnea in convalescent individuals during the first 140 DPO (log-rank test, $P > 0.05$ for VN, $P > 0.05$ for IgG, ****$P < 0.0001$ for IgM).

**Trajectories of antibody titers in subjects who donated plasma more than once.** To determine the kinetics and persistence of IgM, IgG, and VN responses, we next performed longitudinal analyses of the initial and final observed titers in 105 subjects with multiple plasma donations (median 4 donations, interquartile range (IQR): 2–6; median interval between initial and final donation of 42 days (range 6–101 days; IQR: 26–68 days), Supplementary Figs. S1 and S2). The data confirm the robustness of IgG and IgM levels through the 140 DPO observation period. All individuals with a detectable starting titer remained, on average, between one or two dilutions above or below the initial titer (Supplementary Fig. S1). Of particular note, only 5 of 60 individuals (8.3%, 95% CI: 2.8–18.4%) with an initial VN titer of ≥$\log_2$ 5.3 (1:40) had a subsequent increase in titer. This finding emphasizes the importance of recruiting and screening convalescent plasma donors quickly, as VN titers are unlikely to increase from levels at the time donors first become eligible.

## Discussion

The optimal timeframe for donating convalescent plasma to be used for COVID-19 immunotherapy is important to know, since effective treatment outcomes are dependent on the levels of plasma neutralizing antibodies and are independent of whether the individual donor may be protected by a memory or cell-mediated immune response. To address this important knowledge deficit, we determined in vitro live-virus neutralizing capacity and persistence of IgM and IgG antibody responses against the receptor-binding domain and ectodomain of the SARS-CoV-2 spike glycoprotein in 540 convalescent plasma samples obtained from 175 COVID-19 plasma donors for up to 142 DPO. The persistence of IgG responses in many convalescent individuals through 140 DPO is encouraging from the perspective of antibody durability to SARS-CoV-2 and are indicative of a robust response to infection in the majority of individuals with RT-PCR confirmed infection (Fig. 1C–F). These findings are consistent

with the expected serological responses to rapidly replicating RNA viruses, including SARS-CoV-1 and MERS-CoV[20–22].

In contrast, the persistence of IgM well beyond the acute phase was unexpected and differs from reports suggesting a rapid decline in IgM by 4–6 weeks[20,23]. Persistence of IgM in circulation beyond the more typical acute phase response period for SARS-CoV-2 infection is reported from other studies[10,24]. Long-term persistence of IgM in circulation has also been noted for other rapidly replicating RNA viruses such as influenza[25]. While the precise mechanisms driving long-term IgM persistence for SARS-CoV-2 are unknown, these may include factors such as chronic or persistent antigen stimulation of naïve B cells that differentiate into IgM-secreting cells, constitutive IgM production by natural IgM-secreting B-1a cells, stimulation of memory B-1b cells elicited during primary infection by antigen re-exposure, or maintenance of IgM production by long-term IgM-secreting plasmablasts or mature plasma cells in the spleen or bone marrow[26]. Regardless of mechanism, IgM persistence up to 140 DPO has important implications for the diagnosis of recent infection based on the presence of IgM alone. Determining the role and contribution of such persistence to pathology and protection remains a promising area of future research. To explore the potential role of IgM in neutralization, IgM depletion studies on a small number of samples ($n = 9$) were conducted (Supplementary Fig. S5). Briefly, the results show that DTT treatment of plasma results in a notable reduction in S/RBD IgM ELISA titer but not S/RBD IgG or in vitro live virus neutralization titers. While the VN titers were unchanged (ID# 0368, DPO 29) on DTT treatment or reduced by, on average, $\log_2$ 1.33 +/− 0.29, it is noteworthy that some samples (e.g., #0016, DPO 32) showed a larger reduction in titer of at least 8-fold (640 to <80), suggesting that the removal of IgM with DTT treatment is likely to have a variable response in different samples. These results, in agreement with other reports[27,28], suggest a role for both IgG and IgM in neutralization observed in live virus in vitro neutralization assays for SARS-CoV-2. Future studies should be conducted to determine the exact contribution of IgG, IgM, IgA, and other isotypes to in vitro neutralization of SARS-CoV-2 and if these parameters vary between individuals and over time.

Antibodies directed against SARS-CoV-2 S/ECD and S/RBD neutralize the virus in vitro, and several vaccines targeting the S glycoprotein have shown promise in animal infection models and human clinical trials[29–33]. We have previously reported that anti-S/RBD or anti-S/ECD antibody titers of ≥1350 are strong proxies for a VN titer ≥160, the value the FDA recommended for use in COVID-19 convalescent plasma therapy prior to August 2020[12,34]. More importantly, we and others have reported that transfusion of anti-S/RBD IgG ≥1350 titer plasma within 72 h (h) of hospitalization significantly improves survival and health outcomes[8,9], and that transfusion of convalescent plasma with a high VN titer of ≥160 resulted in good recovery of patients[35,36] as compared to patients who received plasma with low VN titers[37,38]. However, the guidance for high-titer convalescent plasma suitable for transfusion is still evolving, and additional studies are needed to benchmark the suitable VN or ELISA antibody titers to use for plasma therapy[7].

A theoretical risk of anti-S glycoprotein antibodies is antibody-dependent enhancement (ADE). ADE can occur when non-neutralizing antibodies or antibodies at sub-neutralizing levels bind to viral antigens without blocking or clearing infection[39]. No definitive role for ADE in human coronavirus infections has been established, and whether a threshold level of VN titer that may trigger ADE exists requires further investigation. Importantly, the prevailing evidence suggests that even lower titer convalescent plasma does not trigger ADE[32].

The current study suggests that an S/RBD ≥1350 titer is a promising marker for identifying suitable plasma donors with high virus neutralizing antibody titers early (within 60 days), but

not late, after first symptom onset. In contrast, we find that S/RBD IgG ≥1350 appears to be a reliable predictor of VN response since S/RBD IgG ≥1350 survival curves are statistically indistinguishable from those of VN ≥ 160, with robust positive and negative likelihood ratios suggesting that isotype-specific IgG assays may be better predictors than those that are based on the Fab fragment. The exact reasons for these differences in predictive value between anti-S/RBD Fab-based total antibody titers and isotype-based assays are unknown and need further investigation.

Our findings of robust predictive values of S/RBD IgG titers IgG ≥1350 as proxies for VN ≥ 160 are especially relevant given the mounting need for facile methods to identify suitable convalescent plasma donors are needed as the gold standard live-virus VN assays used herein are labor intensive, cumbersome, take several days to perform, and require specialized expertise and access to a biosafety level 3 laboratory and regulatory approvals. Surrogate assays using virus pseudotypes have been developed and standardized, which can be tested at BSL2 level[40–43]. Assays have been developed to quantify neutralizing antibodies that block binding of SARS-CoV-2 RBD to human ACE-2 receptor[44]. However, these surrogate assays have similar drawbacks to conventional neutralization assays except that they can be performed in BSL2 facilities. These assays are expensive, have limited supply of reagents and difficult to harmonize when applied on a larger scale to screen populations. Finally, we and others have used antibody titers against spike and RBD domains of SARS-CoV-2 virus that have been shown to correlate well with the VN assays[12,45,46]. ELISAs are easier to implement than VN or surrogate VN assays, especially in resource-limited countries and environments.

The observation that the proportion of individuals with a VN titer ≥160 remained above 80% through the first 60 DPO (Fig. 2C, D), indicate that the time period in which donated convalescent plasma is likely to have a high VN titer and optimal therapeutic potential is within the first 60 DPO. Together with the observation that asymptomatic and mildly infected individuals mount less robust serological responses than individuals with severe infection who are more likely to become eligible for plasma donation at later time points post onset of symptoms, these findings suggest there is a relatively narrow window for donation of high-titer convalescent plasma for use in immunotherapy for COVID-19 patients that begins to close within two months of symptom onset. This result has important implications for convalescent plasma donation and passive immunotherapy programs, some of which have already transfused nearly 95,000 individuals in the United States as of December, 2020 (https://www.uscovidplasma.org), and especially so given the demographic shift to a younger age group with mild or asymptomatic infections who appear less likely to mount robust virus neutralizing serological responses.

Recently there has been increasing evidence that IgA is an important contributor to the virus neutralizing response against SARS-CoV-2[47,48]. We have not quantified the IgA responses in the donor cohort and is an important limitation of our study. It is important to note here that there may be potential confounding of plasma IgA in Fab fragment determination. Also, we have only quantified the antibody titers toward either spike or RBD domain of SARS-CoV-2; epitopes present on other proteins, including nucleocapsid, may also contribute to the functional neutralizing antibody titers[49,50]. Finally, it is noted that studies on functional aspects of the antibodies with relation to their virus neutralizing titers is an important area for future research.

To summarize, our findings refine our understanding of the kinetics, magnitude, and durability of human serologic responses to SARS-CoV-2 spike protein, the primary vaccine candidate being studied worldwide. This integrative analysis suggests that while robust and persistent live virus VN and serological response to SARS-CoV-2 S/ECD and S/RBD, there is a limited donation

window of ~60 DPO for high-titer anti-spike protein convalescent plasma suitable for immunotherapy for COVID-19 patients. Together, these findings define the optimal window for donating convalescent plasma useful for immunotherapy of COVID-19 patients and reveal important predictors of an ideal plasma donor.

## Methods

**Cohort and sample description.** Plasma samples ($n = 540$) from 175 COVID-19 convalescent patients collected at Houston Methodist Hospital in Houston, Texas were studied. Patients were confirmed to be positive for SARS-CoV-2 by RT-PCR. The severity of infection in these patients was scored on a scale of 1–5, (median 2, IQR: 1–2). A score was assigned to the patients on an ordinal scale of disease severity as follows: 1 = mild disease without dyspnea; 2 = moderate disease with dyspnea that did not require supplemental oxygen or hospitalization but requiring medical care; 3 = moderate disease with dyspnea that required low-flow supplemental oxygen and hospitalization; 4 = severe disease that required supplemental oxygen through non-invasive ventilation or high-flow oxygen devices post hospitalization; 5 = critical disease that required intensive care unit admission and invasive mechanical ventilation or extracorporeal membrane oxygenation (ECMO).

Per FDA guidelines (https://www.fda.gov/vaccines-blood-biologics/investigational-new-drug-ind-or-device-exemption-ide-process-cber/recommendations-investigational-covid-19-convalescent-plasma#Patient%20Eligibility), all subjects were asymptomatic for at least 14 days at the time of plasma collection. Of the 175 subjects, 105 individuals donated convalescent plasma at least twice (range 2–12 times) (Supplementary Figs. S1 and S2). All donors were confirmed negative for SARS-CoV-2 by RT-PCR and provided written consent before plasmapheresis. The study cohort consisted of 88 females (50.3%) and 87 males (49.7%), ranging in age between 20 and 78 years (median 46, IQR: 36–54). Samples were collected from 17–142 DPO (median 68 DPO, IQR: 48–93). Plasma from donors was collected with an apheresis system (Trima Accel® Terumo BCT) and standard blood banking protocols were followed. An aliquot of collected plasma was tested for antibodies by ELISA and/or VN assays. Cohort characteristics are described in Table 1 and Supplementary Table S1.

**Quantitative estimation of antibodies against SARS-CoV-2.** SARS-CoV-2 antibodies in plasma samples were detected and quantified against purified recombinant SARS-CoV-2 spike ectodomain (S/ECD) or receptor-binding domain (S/RBD) proteins using in-house indirect Fab antibody-based or isotype-specific (IgM and IgG) ELISA assays. The protocols are deposited in protocols.io (dx.doi.org/10.17504/protocols.io.bivgke3w)[12,51]. Two isotypes of CR3022, a human monoclonal antibody reactive to spike regions of SARS-CoV-1 and SARS-CoV-2, were used as positive controls in the assays (IgG1: Ab01680-10.0; IgM: Ab01680-15.0, Absolute Antibody, USA). The cutoff for the assays was determined as an optical density (absorbance at 450 nm) higher than three or six standard deviations above the mean of the tested pre-COVID-19 serum samples ($n = 100$). Sample titers were estimated as reciprocals of the highest dilution resulting in an OD greater than the cutoff. The class specificity of the IgM ELISA was tested by treating the plasma samples ($n = 10$) with 1,4-dithiothreitol (DTT, 10708984001, Millipore Sigma, USA)[52]. Briefly, samples were allowed to react with 0.005 M DTT in phosphate buffered saline (PBS, pH 7.4) at $36 \pm 2\,°C$ for 30 min and then tested with isotype-specific ELISAs for titer estimation (Supplementary Fig. S6).

**Virus neutralization assay.** The VN titers of the plasma samples were quantified on a cell-based assay using SARS-CoV-2 strain USA-WA1/2020 (NR-52281-BEI Resources, USA)[12,53]. Briefly, Vero E6 cells (CRL-1586, ATCC, USA) were grown as monolayers in 96-well microtiter plates. Heat-inactivated plasma samples were diluted twofold in triplicate and incubated with 100 tissue culture infective dose 50 ($TCID_{50}$) of the virus at 5% $CO_2$ at $36 \pm 2\,°C$ for 60 min. This plasma–virus mixture was added to cell monolayers and incubated further for 72 h at 5% $CO_2$ at $36 \pm 2\,°C$. Plates were treated with crystal violet formaldehyde stain for 1 h and visually inspected for cytopathic effect (CPE) or protection. The reciprocal of the highest dilution of the plasma where at least two of the three wells were protected (no CPE) was determined as the VN titer of the sample.

**IgM depletion assay.** IgM was depleted from plasma samples ($n = 9$) with 0.005 M DTT as described above. DTT was neutralized with 0.025 M iodoacetamide (I1149, Millipore Sigma, USA). The treated plasma samples were dialyzed against PBS, pH 7.4 for 12–18 h at 4 °C. The samples were analyzed for S/RBD specific IgM, IgG titers, and VN titers (Supplementary Fig. S5) after dialysis.

**Statistics and reproducibility.** Tests for normality were performed using the Kolmogorov-Smirnov test and a $P$ value of <0.05 was considered statistically significant. Data dispersion was indexed by standard errors of mean or quartile and IQR. The agreement between the various assays was determined using Pearson correlation coefficient with $\log_2$-transformed titers. The non-parametric regression method LOESS was used for scatterplot smoothing to visualize antibody trajectories. The geom_smooth (method = "loess") function in R was used with default span of 0.75.

The proportion of the sample population remaining seropositive over the 100-day period was determined using a log-rank test and Kaplan–Meier survival curves were plotted with "survival" and "survminer" packages in R Studio[54–56]. Statistical differences in antibody titers and survival curves of patient characteristics—including severity score, age, and presence of dyspnea—were analyzed using one-way ANOVAs (Tukey's multiple comparison tests) or unpaired $t$ test, and a log-rank test, respectively. Individual level interval-censored data were used to fit semi-parametric accelerated failure time models using the icenReg R package. DTComPair R package (https://cran.r-project.org/web/packages/DTComPair/DTComPair.pdf) was used to compare the sensitivity, specificity, and positive and negative predictive values for detection of S/RBD, S/ECD, and S/RBD IgG titers ≥1350, as well as S/RBD IgM titer ≥450 using VN titer ≥160 as the gold standard. Positive and negative predictive values were compared with the generalized score statistics, whereas the sensitivity and specificity were compared using an exact binomial test. All analyses were completed using R (versions 3.6.1 or 3.6.3) within R Studio (version 1.2.5019) or Graphpad PRISM 8 (version 8.4.3). The samples were assayed either in duplicates or triplicates and the results were tested for reproducibility with randomly selected samples.

**Study approvals.** Informed consent was obtained from either the patient or an authorized representative of the patient when applicable for collection of plasma samples. All procedures were approved by the Institutional Review Board of Houston Methodist Hospital (IRB# PRO00025121). Serological analyses were performed at the Pennsylvania State University under BSL-2 (ELISA assays) and BSL-3 (VNs) conditions, following the Pennsylvania State University Institutional Biosafety Committee (IBC) approved protocols.

**Reporting summary.** Further information on research design is available in the Nature Research Reporting Summary linked to this article.

## Data availability

All data generated or analyzed during this study are included in this published article (and its supplementary information files) or will be made available by the authors on reasonable request.

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

## Acknowledgements

We are deeply indebted to all of our volunteer plasma donors for their time, their generous gift, and their solidarity. We thank Katharine G. Dlouhy, Curt Hampton, and their team of coordinators and recruiters for outstanding efforts; and Monisha Dey, Cheryl Chavez-East, John Rogers, Dr. Ahmed Shehabeldin, Dr. David Joseph, Guy Williams, Karen Thomas, and Curt Hampton who were instrumental in efficiently managing the donor center; Drs. Jessica Thomas and Zejuan Li, Erika Walker, the very talented and dedicated molecular technologists, and the many labor pool volunteers in the Molecular Diagnostics Laboratory for their dedication to patient care; the many donor center and blood bank phlebotomists and technologists for their dedication to donor and blood safety; Drs. Heather McConnell and Sasha M. Pejerrey for outstanding editorial assistance; Brandi Robinson, Harrold Cano, and Cory Romero for technical assistance; Claude Moussa, Heather Patton, and the many members of the laboratory information technology team for rapidly implementing the necessary electronic workflows; Pamela McShane, Dilzi Mody, and the many members of the biorepository team for their meticulous management of patient samples; and Christina Talley, Dr. Susan Miller, and Mary Clancy for consistent, thorough, and outstanding advice. We express our gratitude to Manuel Hinojosa and Mark Vassallo for their extensive efforts to rapidly procure resources, and Dr. Roberta Schwartz for her efforts in implementing screening of asymptomatic individuals. We are indebted to Drs. Marc Boom and Dirk Sostman for their support, and to many very generous Houston citizens and businesses for their tremendous philanthropic support of this ongoing project, including but not limited to anonymous, Ann and John Bookout III, Carolyn and John Bookout, Ting Tsung and Wei Fong Chao Foundation, Ann and Leslie Doggett, Freeport LNG, the Hearst Foundations, Jerold B. Katz Foundation, C. James and Carole Walter Looke, Diane and David Modesett, the Sherman Foundation, Paula and Joseph C. "Rusty" Walter III, and Aramco Americas. Dr. Jason S. McLellan (University of Texas at Austin) graciously provided the mAb CR3022 and the spike protein expression vectors, and we thank the members of the Center for Systems and Synthetic Biology at the University of Texas at Austin for technical assistance. We thank Zivko Nikolov, Susan Woodard, and Michael Johanson at the National Center for Therapeutics Manufacturing at Texas A&M University for production of antigen. We thank Terumo BCT for continuously and rapidly supplying blood collection devices and supplies, and Victoria Cavener and Team COVID-19 serology at Penn State for their timely and generous technical assistance and logistical support. This study was supported by the Fondren Foundation, Houston Methodist Research Institute (to J.M.M.). A portion of this work was funded through Cooperative Agreement W911NF-12-1-0390 by the Army Research Office (to J.G.). We gratefully acknowledge the unwavering support and timely seed funding from the Huck Institutes of the Life Sciences for the studies at Penn State together with the Huck Distinguished Chair in Global Health award (to V.K.).

## Author contributions

Project concept (A.G., S.S., E.S., S.V.K., J.M.M., and V.K.); acquired data (A.G., S.S., E.S., M.S.N., R.H.N., D.G., I.M.B., I.P., R.K., S.E.L., A.M.M., R.R., P.A.C., B.C., J.C., T.N.E., X.Y., P.Z., C.L., R.J.O., D.W.B., and J.G.); analyzed data (A.G., S.S., M.S.N., C.H., M.J.F., S.V.K., J.M.M., and V.K.); wrote manuscript (V.K., J.M.M., A.G., S.S., and S.V.K.); prepared figures (A.G., S.S., M.S.N., C.H., and V.K.). All authors reviewed the manuscript and gave final approval for publication.

## Competing interests

E.S. is the local principal investigator for a clinical trial sponsored by Regeneron assessing an investigational therapy for COVID-19.
