## [Peer Review File · Communications Biology]

Reviewers' comments:

Reviewer #1 (Remarks to the Author):

In this article, Gontu and colleagues report plasma spike-protein (RBD and ECD) antibody and neutralizing titers in convalescent plasma obtained from patients who recovered from COVID-19. Their results show that RBD and ECD IgG and IgM largely persisted over a period of more than 100 days post infection, whereas SARS-CoV-2 neutralizing titers (measured by live virus assay) declined as soon as 60 days post symptoms and conclude their findings " ... define the optimal window for donating convalescent plasma for immunotherapy of COVID-19 patients" (lines 50-52).

The study is well done and provides an important longitudinal snapshot of a population based survey of the spike protein antigen and neutralizing antibody responses of plasma donors ranging in age from their 20's to > 60 years who had recovered from different degrees of COVID-19 disease severity. In many ways, the results mirror other studies, with the caveat that the persistence of antibody appears to differ, perhaps as a function of the antigen assayed and the platform used.

This study provides new and important information. First, it shows that RBD IgM persisted in ~ 60% of the cohort examined for at least 100 days after symptom onset. Second, it shows that an RBD IgG titer of ≥ 1350 is highly predictive of a neutralizing titer of $\geq 1:160$.

The following questions and comments are offered for the authors' consideration:

1. Was IgA measured? If not, why not? Could it be confounding the Fab fragment determinations?
2. What happens to neutralizing activity if IgM is depleted (or if IgG and IgA are depleted)? What happens if IgM is depleted from samples obtained earlier and later after symptom onset?
3. Were other antibody functional activities measured?
4. Were there sex differences in spike protein or neutralizing titers?
5. How many persons' samples were studied at two or more times?
6. The assay the authors utilize is an in-house assay. Does RBD IgG on this assay correlate with other RBD or spike-protein assays, e.g. commercial platforms like OrthoV, which was used to analyze samples from the Mayo Clinic expanded access program study?
7. Can age and disease severity be separated as factors influencing the robustness or persistence of antibody responses?
8. The abstract states that a neutralizing titer of $\geq 1:160$ is "... a value suitable for convalescent plasma therapy." This statement implies this is an established dose. To this reviewers' knowledge, a titer $\geq 1:160$ is recommended based on convalescent plasma treatment of SARS but has not been validated for COVID-19. Please see: Food and Drug Administration Investigational COVID-19 convalescent plasma-emergency INDs http://natap.org/2020/COVID/032320_39.htm and <https://www.fda.gov/vaccines-blood-biologics/investigational-new-drug-ind-or-device-exemption-ide-process-cber/recommendations-investigational-covid-19-convalescent-plasma> and update with new information if available.

Reviewer #2 (Remarks to the Author):

In this paper authors presented data supporting that a limited window for donation of convalescent plasma with high live-virus neutralizing antibodies could be detected and used for COVID-19 immunotherapy. Distribution of antibody titers against SARS-CoV-2 were correlated with age, severity scores, and presence of dyspnea.

Data are clearly presented and the methodology is appropriate.

However, data on race, sex are lacking: these parameters may influence the extent of antibody production and may be of interest for the reader. Further lab parameters may be related to the degree of antibody titers against SARS-CoV-2, and these may be mentioned and discussed in the discussion section (see references listed above)

The reference list should be expanded, accordingly . A systematic review was recently published highlighting several hot topics in this setting (Valk_SJ, Piechotta_V, Chai_KL, Doree_C, Monsef_I, Wood_EM, Lamikanra_A, Kimber_C, McQuilten_Z, So-Osman_C, Estcourt_LJ, Skoetz_N. Convalescent plasma or hyperimmune immunoglobulin for people with COVID-19: a rapid review. Cochrane Database of Systematic Reviews 2020, Issue 5. Art. No.: CD013600.DOI: 10.1002/14651858.CD013600.)

Seghatchian J, Lanza F. Convalescent plasma, an apheresis research project targeting and motivating the fully recovered COVID 19 patients: A rousing message of clinical benefit to both donors and recipients alike. Transfus Apher Sci. 2020 Jun;59(3):102794. doi: 10.1016/j.transci.2020.102794.

Arturo Casadevall, Liise-anne Pirofski. The convalescent sera option for containing COVID-19 J Clin Invest. 2020;130(4):1545-1548. <https://doi.org/10.1172/JCI138003>.

Lanza F, Seghatchian J. Reflection on passive immunotherapy in those who need most: some novel strategic arguments for obtaining safer therapeutic plasma or autologous antibodies from recovered COVID-19 infected patients. Br J Haematol. 2020 May 14:10.1111/bjh.16814. doi: 10.1111/bjh.16814.

Reviewer #3 (Remarks to the Author):

The manuscript Gontu et al details the decay in binding and neutralizing antibody responses in a cohort of 175 subjects sampled longitudinally. They identify a window (<60DPO) as the optimal time to harvest convalescent plasma (CP) for use as a possible therapeutic for COVID-19. The study is well performed, while findings around decay of antibody and neutralizing titres are in line with numerous other reports in this area, the large cohort analysed is a real strength.

1 – the authors have taken a holistic approach to the modelling by combining all subjects and data points. It would be informative to see some longitudinal data stratified by individual presented (first-last analysis in the supplemental gives no indication as the change in the rates of decay within each individual.

2 – In figure 2A, the decay in VN tracks more closely with IgM responses than with IgG titres, that were well maintained. This is also seen in the correlation analysis (1B) where IgM responses correlated best with VN. Is the implication then that IgM underpins most serum neutralization? I feel this should be demonstrated experimentally, with either enrichment/depletion studies in the VN assay, or maybe selective utilization of the DTT treatment could be employed.

3 – While FDA and other have provided cutoffs guidelines for the use of CP therapy, there is not widespread uptake of standardized assays to assess these titres. Can the authors discuss the merits of the microneutralization assay they employed versus others, and comment on the cross-comparability and standardization of these assays as it seems highly relevant for assessing the results of CP trials.

4 – The discussion should be expanded to take in the recent PLACID RCT (reported in BMJ) which, showed limited benefit for plasma therapy (albeit with a lower donor cutoff of 1:80).

5 – I wonder if the authors could similarly discuss the relative risks of ADE in the recruitment of CP donors with strong vs weak neutralizing responses.

COMMSBIO-20-2810-T. Gontu et al. 2020. Response to Reviewers' comments:

We thank the reviewers for their careful review of the manuscript and the many helpful suggestions for improving the study have now been incorporated into the revised manuscript.

Specifically, in response to comments by Reviewers #1 and #3 regarding role of IgM in virus neutralization, we performed additional studies to selectively deplete IgM in a subset of plasma samples to assess impact on live virus neutralization assays. The results are consistent with the hypothesis that both IgG and IgM contribute to *in vitro* live virus neutralization, but their relative contributions are likely to require future detailed investigations.

Detailed responses to individual reviewer comments are provided below:

Reviewer #1 (Remarks to the Author):

In this article, Gontu and colleagues report plasma spike-protein (RBD and ECD) antibody and neutralizing titers in convalescent plasma obtained from patients who recovered from COVID-19. Their results show that RBD and ECD IgG and IgM largely persisted over a period of more than 100 days post infection, whereas SARS-CoV-2 neutralizing titers (measured by live virus assay) declined as soon as 60 days post symptoms and conclude their findings “... define the optimal window for donating convalescent plasma for immunotherapy of COVID-19 patients” (lines 50-52).

The study is well done and provides an important longitudinal snapshot of a population based survey of the spike protein antigen and neutralizing antibody responses of plasma donors ranging in age from their 20's to > 60 years who had recovered from different degrees of COVID-19 disease severity. In many ways, the results mirror other studies, with the caveat that the persistence of antibody appears to differ, perhaps as a function of the antigen assayed and the platform used.

This study provides new and important information. First, it shows that RBD IgM persisted in ~ 60% of the cohort examined for at least 100 days after symptom onset. Second, it shows that an RBD IgG titer of ≥ 1350 is highly predictive of a neutralizing titer of $\geq 1:160$.

The following questions and comments are offered for the authors' consideration:

1. Was IgA measured? If not, why not? Could it be confounding the Fab fragment determinations?

IgA was not measured in this study. While our manuscript was under review, it has been shown that IgA is likely to contribute to the early neutralizing antibody responses to SARS-CoV-2 (DOI: 10.1126/scitranslmed.abd2223). We acknowledge this limitation in the Discussion section of the revised manuscript (Lines 343-346). Further, the potential confounding of plasma IgA in Fab fragment determination is also acknowledged (Lines 343-346).

2. What happens to neutralizing activity if IgM is depleted (or if IgG and IgA are depleted)? What happens if IgM is depleted from samples obtained earlier and later after symptom onset?

We agree that these are important questions that warrant further investigation; however, such investigation is perhaps beyond the scope and focus of the current study. That being said, to begin assessing the impact of IgM depletion on live virus neutralization, we performed preliminary studies on nine plasma samples with high VN titers to selectively deplete IgM using DTT. These method is now described in lines 418-422 of the revised manuscript.

In brief, DTT-treated plasma samples were dialyzed against PBS for 12-18 hrs at 4°C. IgM and IgG ELISAs were performed on both untreated and DTT-treated samples to confirm depletion of IgM before assessment of live SARS-CoV-2 VN titers.

The table below summarizes our results (described in lines 272-283 of the revised manuscript and included as supplemental figure 5):

ID	DPO	S/ RBD IgG		S/ RBD IgM		VN	
		Plasma	DTT treated	Plasma	DTT treated	Plasma	DTT treated
0016	32	4050	4050	1350	0	640	80*
0879	34	4050	1350	4050	50	320	80*
0464	118	1350	1350	450	50	160	80*
3063	93	1350	450	150	50	160	80*
0838	98	4050	450	150	50	640	320
0598	99	4050	1350	150	0	320	160
0620	40	4050	4050	4050	50	640	160
0262	99	4050	4050	1350	0	160	80
0368	29	4050	450	50	50	320	320

The results show that DTT treatment of plasma results in a notable reduction in S/RBD IgM ELISA titer but not S/RBD IgG or in vitro live virus neutralization titers. While the VN titers were unchanged (e.g. ID# 0368, DPO 29) on DTT treatment or reduced by, on average, $\log_2 1.33 \pm 0.29$, it is noteworthy that some samples (ID# 0016, DPO 32) showed a larger reduction in titer (640 to <80), suggesting that removal of IgM with DTT treatment is likely to have a variable response in different samples. For a subset of samples (ID# 0016, 0879, 0464, 0838), the lowest detectable endpoint in VN assays was only 80 because of non-specific CPE likely resulting from residual DTT and iodoacetamide, and hence only approximate reductions in VN can be made. Importantly, these observations are in agreement with other recent reports (DOI: 10.1101/2020.10.09.333278) that suggest a role for both IgG and IgM in neutralization observed during live virus in vitro VN for SARS-CoV-2. Future studies should be planned to determine the exact contribution of IgG, IgM, IgA and other isotypes to in vitro neutralization of SARS-CoV-2 and how this may vary between individuals and over time as is also noted in the revised manuscript (lines 272 - 283). As noted above, we believe these extensive studies are well outside the scope and focus of this manuscript.

Finally, since the plasma samples were collected at least 14 days after a negative PCR result for use in convalescent plasma therapy and not during the very early phases of infection, future studies should also investigate the role of IgM on VN titers in samples taken very early versus during later stages of infection as well as from individuals in different age groups and those with varying severity of symptoms. Although important to investigate, again, we believe these extensive studies are well outside the scope and focus of this manuscript.

3. Were other antibody functional activities measured?

Antibody functions other than binding and virus neutralization were not measured in this study. Our focus primary focus was to better understand the persistence of IgG, IgM and neutralizing antibody titers in convalescent plasma, and the risk factors associated with persistence. Future work should focus on studying other functional activities of antibodies in regards to SARS-CoV-2. This is now noted in the manuscript (Lines 349-350).

4. Were there sex differences in spike protein or neutralizing titers?

As shown in the adjacent figure, statistically significant differences were observed between titers in males and females for S/RBD IgG. However, no differences were observed in titers between males and females for S/RBD IgM and VN. This figure has now been included in the supplementary materials Supplementary Figure 3).

5. How many persons' samples were studied at two or more times?

A total of 105 individuals donated samples two or more times. This is noted on lines #379-380.

6. The assay the authors utilize is an in-house assay. Does RBD IgG on this assay correlate with other RBD or spike-protein assays, e.g. commercial platforms like OrthoV, which was used to analyze samples from the Mayo Clinic expanded access program study?

Yes, the SARS-CoV-2 S/ECD and S/RBD ELISA performed at HMH were benchmarked with the OrthoV assay as has been previously reported in <https://doi.org/10.1016/j.ajpath.2020.10.008>. In brief, the results show a strong positive correlation between the S/RBD IgG and the Ortho VITROS IgG test for 1142 samples with parallel assessment ($R = 0.88$; $P < 0.001$). We have also benchmarked the in-house assay with the EuroImmune assay and show a Pearson correlation of 0.69 (95% CI: 0.65, 0.73) amongst 600 samples run in parallel, together suggesting the in-house S/RBD IgG ELISA is well correlated with commercial assays.

7. Can age and disease severity be separated as factors influencing the robustness or persistence of antibody responses?

*Our explanation and answer to the above question can be found on lines 197-202. **As described in lines 193-198:** Age and severity score were only weakly correlated (Spearman rank correlation=0.08; $P=.07$), but formal analysis of confounding or interactions between age and severity was precluded due to data frailty and requires further study." Regardless, our findings suggest that convalescent individuals <30 years of age and those with lower disease severity scores are less likely to be suitable donors of convalescent plasma for immunotherapy*

for COVID-19 patients than individuals in >30 age group with a history of more severe disease. See also our earlier study <https://doi.org/10.1172/JCI141206>.

8. The abstract states that a neutralizing titer of $\geq 1:160$ is "... a value suitable for convalescent plasma therapy." This statement implies this is an established dose. To this reviewers' knowledge, a titer $\geq 1:160$ is recommended based on convalescent plasma treatment of SARS but has not been validated for COVID-19. Please see: Food and Drug Administration Investigational COVID-19 convalescent plasma—emergency INDs http://natap.org/2020/COVID/032320_39.htm and <https://www.fda.gov/vaccines-blood-biologics/investigational-new-drug-ind-or-device-exemption-ide-process-cber/recommendations-investigational-covid-19-convalescent-plasma> and update with new information if available.

The reviewer rightly notes that the initial FDA-recommended VN titer of ≥ 160 for convalescent plasma therapy that was in place prior to August 2020 appears to have been a somewhat arbitrary value and is still evolving. For instance, the current FDA recommendations are for a neutralizing antibody titer of ≥ 250 in the Broad Institute's neutralizing antibody assay, a signal-to-cutoff (S/C) of ≥ 12 in the Ortho VITROS IgG assay, or a level of $\geq 1:2880$ in the Mount Sinai COVID-19 ELISA IgG Antibody Test; and these units are labeled as "High Titer COVID-19 Convalescent Plasma" or "High Titer CCP". Units with lower titers may be labeled as "COVID-19 Convalescent Plasma of Low Titer" and used if high titer units are not available. Prior to August 2020, titers of 1:160 or even 1:80 using a variety of different assays were considered acceptable if higher-titer units were not available.

Important to note however, although the guidance from the FDA is still evolving, there is accumulating evidence that a VN titer of ≥ 160 resulted in good recovery of patients as compared to patients who received plasma with low VN titers (DOI: 10.1001/jama.2020.10044, 10.3324/haematol.2020.261784, 10.3324/haematol.2020.261784). However, further validation is necessary to benchmark the suitable VN or ELISA antibody titers for plasma therapy. Verbiage in the Abstract has been modified accordingly. This has now been discussed in lines 287-296.

Reviewer #2 (Remarks to the Author):

In this paper authors presented data supporting that a limited window for donation of convalescent plasma with high live-virus neutralizing antibodies could be detected and used for COVID-19 immunotherapy. Distribution of antibody titers against SARS-CoV-2 were correlated with age, severity scores, and presence of dyspnea. Data are clearly presented and the methodology is appropriate.

However, data on race, sex are lacking: these parameters may influence the extent of antibody production and may be of interest for the reader. Further lab parameters may be related to the degree of antibody titers against SARS-CoV-2, and these may be mentioned and discussed in the discussion section (see references listed above)

The reference list should be expanded, accordingly. A systematic review was recently published highlighting several hot topics in this setting (Valk_SJ, Piechotta_V, Chai_KL, Doree_C,

Monsef_I, Wood_EM, Lamikanra_A, Kimber_C, McQuilten_Z, So-Osman_C, Estcourt_LJ, Skoetz_N. Convalescent plasma or hyperimmune immunoglobulin for people with COVID-19: a rapid review. Cochrane Database of Systematic Reviews 2020, Issue 5. Art. No.: CD013600.DOI: 10.1002/14651858.CD013600.)

Seghatchian J, Lanza F. Convalescent plasma, an apheresis research project targeting and motivating the fully recovered COVID 19 patients: A rousing message of clinical benefit to both donors and recipients alike. Transfus Apher Sci. 2020 Jun;59(3):102794. doi: 10.1016/j.transci.2020.102794.

Arturo Casadevall, Liise-anne Pirofski. The convalescent sera option for containing COVID-19 J Clin Invest. 2020;130(4):1545-1548. <https://doi.org/10.1172/JCI138003>.

Lanza F, Seghatchian J. Reflection on passive immunotherapy in those who need most: some novel strategic arguments for obtaining safer therapeutic plasma or autologous antibodies from recovered COVID-19 infected patients. Br J Haematol. 2020 May 14;10.1111/bjh.16814. doi: 10.1111/bjh.16814

Thank you for the suggestions. We concur that these references are helpful and provide additional context. The reference list has been updated as recommended.

All the available demographic details for the cohort are now presented in Table 1 of the revised manuscript.

Reviewer #3 (Remarks to the Author):

The manuscript Gontu et al details the decay in binding and neutralizing antibody responses in a cohort of 175 subjects sampled longitudinally. They identify a window (<60DPO) as the optimal time to harvest convalescent plasma (CP) for use as a possible therapeutic for COVID-19. The study is well performed, while findings around decay of antibody and neutralizing titres are in line with numerous other reports in this area, the large cohort analysed is a real strength.

1 – the authors have taken a holistic approach to the modelling by combining all subjects and data points. It would be informative to see some longitudinal data stratified by individual presented (first-last analysis in the supplemental gives no indication as the change in the rates of decay within each individual.

Thank you, we agree. We had initially included the first and last titers to provide the overall trajectories of antibody decay with time. Rates of decay within each individual are now shown as well in the revised supplemental section (Fig. S2).

2 – In figure 2A, the decay in VN tracks more closely with IgM responses than with IgG titres, that were well maintained. This is also seen in the correlation analysis (1B) where IgM responses correlated best with VN. Is the implication then that IgM underpins most serum neutralization? I feel this should be demonstrated experimentally, with either enrichment/depletion studies in the VN assay, or maybe selective utilization of the DTT treatment could be employed.

Thank you. We have revised the manuscript according to your suggestions.

We agree that these are important questions that warrant further investigation; however, such investigation is perhaps beyond the scope and focus of the current study. That being said, to begin assessing the impact of IgM depletion on live virus neutralization, we performed preliminary studies on nine plasma samples with high VN titers to selectively deplete IgM using DTT. These method are now described in lines 418-422 of the revised manuscript.

In brief, DTT-treated plasma samples were dialyzed against PBS for 12-18 hrs at 4°C. IgM and IgG ELISAs were performed on both untreated and DTT-treated samples to confirm depletion of IgM before assessment of live SARS-CoV-2 VN titers.

The table below summarizes our results (described in lines 272-283 of the revised manuscript and included as supplemental figure 5):

ID	DPO	S/ RBD IgG		S/ RBD IgM		VN	
		Plasma	DTT treated	Plasma	DTT treated	Plasma	DTT treated
0016	32	4050	4050	1350	0	640	80*
0879	34	4050	1350	4050	50	320	80*
0464	118	1350	1350	450	50	160	80*
3063	93	1350	450	150	50	160	80*
0838	98	4050	450	150	50	640	320
0598	99	4050	1350	150	0	320	160
0620	40	4050	4050	4050	50	640	160
0262	99	4050	4050	1350	0	160	80
0368	29	4050	450	50	50	320	320

Please see response to reviewer 1, comment 2 and on lines 272 – 283 of the revised manuscript and supplementary figure 5. While the focus and important findings of our studies related to duration of persistence and risk factors associated with decay of antibody titers, we agree with the reviewer that the observations raise important questions regarding the relative contribution of IgM to live virus neutralization. To begin to address this, we performed the DTT treatment for a subset of samples as suggested and also detailed in response to Reviewer 1, Comment 2 above. In brief, the results suggest that both IgG and IgM likely contribute to virus neutralization, and that the future studies are needed to determine the relative contribution of each of these isotypes depending on duration since symptom onset, severity of infection, age, sex, etc.

3 – While FDA and other have provided cutoffs guidelines for the use of CP therapy, there is not widespread uptake of standardized assays to assess these titers. Can the authors discuss the merits of the microneutralization assay they employed versus others, and comment on the

cross-comparability and standardization of these assays as it seems highly relevant for assessing the results of CP trials.

We concur that the development and assessment of standard neutralization assays is critical. We have included a discussion of the pros and cons of the different assays in the Discussion (lines 314-327). Although the live virus neutralization assays used and described in the manuscript are considered as the gold standard assays for assessing the functional neutralizing abilities of antibodies, they require BSL3 laboratories, are labor intensive, time consuming and not widely available. Surrogate assays using virus pseudotypes have been developed and standardized, which can be tested at BSL2 level (DOI: 10.1080/22221751.2020.1743767, <https://doi.org/10.1080/22221751.2020.1815589>). Assays have been developed to quantify neutralizing antibodies blocking the binding of SARS-CoV-2 RBD to human ACE-2 receptor (DOI: 10.1038/s41587-020-0631-z). However, these surrogate assays have similar drawbacks as of the conventional neutralization assays except that these can be performed at BSL2 level. These assays are expensive, have limited supply of reagents and difficult to harmonize when applied on a larger scale to screen populations. Finally, we and others have explored the use of antibody titers against Spike and RBD domains of SARS-CoV2 virus that have been shown to correlate well with the virus neutralization assays (DOI: 10.1172/JCI141206, [10.1007/s15010-020-01503-7](https://doi.org/10.1007/s15010-020-01503-7), [10.1016/j.xcrm.2020.100040](https://doi.org/10.1016/j.xcrm.2020.100040)). ELISAs are easier to implement than VN or surrogate VN assays, especially in resource-limited countries and environments. This is now discussed in the revised manuscript (Lines 314-327).

4 – The discussion should be expanded to take in the recent PLACID RCT (reported in BMJ) which, showed limited benefit for plasma therapy (albeit with a lower donor cutoff of 1:80).

Thank you for the suggestion. We have revised the revised manuscript to discuss clinical trials for plasma therapy with both high (DOI: 10.1001/jama.2020.10044, [10.3324/haematol.2020.261784](https://doi.org/10.3324/haematol.2020.261784)) and low (PLACID RCT: DOI: 10.1136/bmj.m3939) VN titer cut-offs. This has been included in the Discussion section in line no. 289-296.

5 – I wonder if the authors could similarly discuss the relative risks of ADE in the recruitment of CP donors with strong vs weak neutralizing responses.

Thank you for the suggestion. We have revised the manuscript to discuss this point in lines 298-303. A theoretical risk of anti-S glycoprotein antibodies is antibody dependent enhancement (ADE). Briefly, ADE can occur when non-neutralizing antibodies or antibodies at sub-neutralizing levels bind to viral antigens without blocking or clearing infection (<https://doi.org/10.1038/s41564-020-00789-5>). No definitive role for ADE in human coronavirus infections has been established, and whether a threshold level of VN titer that may trigger ADE exists requires further investigation. Importantly, the prevailing evidence suggests that even lower titer convalescent plasma does not trigger ADE (<https://www.bmj.com/content/371/bmj.m3939>).

The FDA recommendation is to qualify units as “high titer” prior to use, thus likely avoiding the theoretical risk of ADE. Since it may not always be feasible to evaluate the VN titers of CP, we recently demonstrated a strong positive correlation between ELISA titers and VN titers. The probability of a VN titer ≥ 160 was 80% or greater with anti-RBD or anti-ECD titers of $\geq 1:1350$ (DOI: 10.1172/JCI141206). This too is now included in the discussion in the revised manuscript (Lines 298-303).

REVIEWERS' COMMENTS:

Reviewer #1 (Remarks to the Author):

The authors were very responsive to the previous review. The data, comments, and references they added strengthen their conclusions and increase the impact of the paper. Although more work is needed - the data in this paper provide a step forward in identifying suitable convalescent plasma donors. On the other hand, it would be helpful to the reader if the authors cited/discussed other studies that have reported similar findings re: decay of antibody responses, IgM neutralization (e.g. Klingler (preprint)) as well as some more recent publications that have found more durable responses (e.g. Wajnberg et al).

Reviewer #2 (Remarks to the Author):

Authors have addressed adequately the issues arisen by reviewers.
This manuscript is now publishable.

Reviewer #3 (Remarks to the Author):

The authors have addressed all my concerns.